# The Impact of Nanobody Density on the Targeting Efficiency of PEGylated Liposomes

**DOI:** 10.3390/ijms232314974

**Published:** 2022-11-29

**Authors:** Bárbara S. Mesquita, Marcel H. A. M. Fens, Alessia Di Maggio, Esmeralda D. C. Bosman, Wim E. Hennink, Michal Heger, Sabrina Oliveira

**Affiliations:** 1Department of Pharmaceutics, Utrecht Institute for Pharmaceutical Sciences, Utrecht University, 3584 CG Utrecht, The Netherlands; 2Cell Biology, Neurobiology and Biophysics, Department of Biology, Science Faculty, Utrecht University, 3584 CG Utrecht, The Netherlands; 3Jiaxing Key Laboratory for Photonanomedicine and Experimental Therapeutics, Department of Pharmaceutics, College of Medicine, Jiaxing University, Jiaxing 314041, China; 4Membrane Biochemistry and Biophysics, Bijvoet Center for Biomolecular Research, Department of Chemistry, Utrecht University, 3584 CG Utrecht, The Netherlands

**Keywords:** biologicals, nanobodies, nanoparticulate drug delivery systems, cancer treatment, protein corona, immune cell interactions

## Abstract

Nanoparticles (NPs) are commonly modified with tumor-targeting moieties that recognize proteins overexpressed on the extracellular membrane to increase their specific interaction with target cells. Nanobodies (Nbs), the variable domain of heavy chain-only antibodies, are a robust targeting ligand due to their small size, superior stability, and strong binding affinity. For the clinical translation of targeted Nb-NPs, it is essential to understand how the number of Nbs per NP impacts the receptor recognition on cells. To study this, Nbs targeting the hepatocyte growth factor receptor (MET-Nbs) were conjugated to PEGylated liposomes at a density from 20 to 800 per liposome and their targeting efficiency was evaluated in vitro. MET-targeted liposomes (MET-TLs) associated more profoundly with MET-expressing cells than non-targeted liposomes (NTLs). MET-TLs with approximately 150–300 Nbs per liposome exhibited the highest association and specificity towards MET-expressing cells and retained their targeting capacity when pre-incubated with proteins from different sources. Furthermore, a MET-Nb density above 300 Nbs per liposome increased the interaction of MET-TLs with phagocytic cells by 2-fold in ex vivo human blood compared to NTLs. Overall, this study demonstrates that adjusting the MET-Nb density can increase the specificity of NPs towards their intended cellular target and reduce NP interaction with phagocytic cells.

## 1. Introduction

Targeted nanoparticles (NPs) have been designed to enhance therapeutic efficacy and reduce the systemic toxicity of different pharmaceuticals and, in particular, anti-cancer agents [1,2,3]. The accumulation of NPs in the tumor is mainly facilitated by the pathophysiological properties of solid tumors, usually characterized by leaky vessels and poor lymphatic drainage; phenomena referred to as the enhanced permeability and retention (EPR) effect [4,5]. When in the tumor area, non-targeted NPs can be internalized indiscriminately by all cells that comprise the tumor microenvironment, i.e., stromal cells and healthy neighboring cells in addition to parenchymal cells. This may limit the therapeutic efficacy of NPs towards the tumor. To increase the specific interaction between the NPs and cancer cells, tumor-targeting moieties that recognize overexpressed proteins on the outer cell membrane (e.g., HER2, MET) can be grafted onto the surface of the NPs. This strategy can lead to higher drug internalization by cancer cells, thereby increasing tumor destruction and reducing off-target tissue damage [6,7,8,9].

The realm of targeting ligands is vast and comprises peptides, antibodies (and their fragments), aptamers, and small molecules (e.g., folic acid and carbohydrates). Monoclonal antibodies (mAbs) are one of the most studied classes of ligands, especially their variable domains, and several antibody-targeting NPs have entered clinical trials [2]. However, full-length antibodies are associated with high manufacturing costs [10] and carry immunological risks even when fully humanized [11]. Conversely, antibody fragments usually have a lower target affinity and are less stable [12]. The discovery of heavy chain-only antibodies in camelids three decades ago [13] led to the exploration of single-domain antibodies, known as VHH or nanobodies (Nbs), as targeting ligands. In contrast to antibody variable fragments, Nbs are smaller (~15 kDa), more resilient against changes in pH and temperature, and have low immunogenicity, which makes them a robust drug-targeting tool [14]. In vitro studies have confirmed the superior uptake and treatment efficacy of NPs (micelles, liposomes, among other types of NPs) decorated with Nbs compared to their non-targeted counterparts [15,16,17,18,19]. Furthermore, core-crosslinked polymeric micelles conjugated with the epidermal growth factor receptor (EGFR)-targeted Nbs reduced EGFR-mediated signaling and inhibited tumor growth in vivo, particularly when loaded with doxorubicin, and outperformed the non-targeted micelles [20].

When designing efficient ligand-targeted NPs, a targeting moiety with high binding affinity and selectivity should be employed along with a suitable conjugation strategy. The latter has been shown to detrimentally affect ligand orientation and conformation [21,22,23]. For example, PEG-PLGA NPs conjugated to the strained alkyne-modified cetuximab Fab (via azide-alkyne copper-free click chemistry) have proven more efficient in EGFR binding than those conjugated to cetuximab with random lysine modification (exploiting carbodiimide chemistry). As antibodies have multiple lysine residues on their surface, random conjugation through this amino acid can disrupt the protein 3D structure and result in a loss of function [22]. Ligand density has also been found to govern targeting efficiency. Excessively covered surfaces can increase steric hindrance and decrease the stealth properties of the NPs [24,25]. PEGylation has been used as a molecular mask to reduce NP opsonization and fast clearance by the mononuclear phagocyte system. Yet, the presence of targeting moieties can obviate the stealth properties and compromise circulation time in the blood and tumor accumulation, as reported for polymeric micelles grafted with RGD peptides and liposomes conjugated with 25 kDa antibody fragments targeting prostate-specific membrane antigen or folate [25,26,27].

In terms of optimal ligand density, some lessons can be learned from viruses. Viruses, like nanomedicines, aim to avoid immune recognition, and the density and distribution of spike proteins on the virion envelope is one of the mechanisms utilized. The HIV virus has a low spike density (on average 14 spikes per virion), whereas lassa and influenza viruses have 270 and 300–400 spikes per virion, respectively, while having the same particle size of around 100 nm [28,29,30]. The dearth of spikes in HIV virions potentially serve as one of the means of immune evasion while retaining efficient receptor binding and viral replication potential [29]. Analogously, the deleterious effects of active targeting on the pharmacokinetics of NPs might be (partially) offset by adjusting the number of ligands per NP [31].

Targeted NPs face another obstacle once they reach the tumor site. The biomolecules and particularly plasma proteins that might have been adsorbed onto their surface in the circulation (the antifouling PEG chains fail to completely prevent protein adsorption [32]) can hide the ligand, preventing receptor-mediated association with the target cells. For instance, transferrin-functionalized silica nanoparticles in the presence of increasing concentrations of serum proteins were reported to have a lower binding affinity towards the cognate receptor and consequently exhibited decreased uptake by A549 cells. This effect has been attributed to the protein corona [33].

This study therefore aimed to elucidate the relationship between Nb density and targeting efficacy of PEGylated liposomes towards cancer cells as well as liposome association with immune cells. PEGylated liposomes were conjugated with a varying number of Nbs raised against the hepatocyte growth factor receptor (MET) (previously described as G2 Nbs [34,35]) via maleimide-thiol click chemistry. MET was chosen as a target because it is a membrane receptor that is frequently overexpressed in numerous tumors, including breast cancer [36], cholangiocarcinoma [37], and glioblastoma [38]. Moreover, several anti-MET Abs with an inhibitory effect and anti-MET antibody-drug conjugates are undergoing clinical trials [39,40,41], which attests to the importance of exploring this protein as a target for drug delivery. MET-targeted liposomes (MET-TLs) and non-targeted liposomes (NTLs) were characterized by dynamic light scattering (DLS) and nanoparticle tracking analysis (NTA), and the extent of particle functionalization with MET-Nbs was assessed by SDS-PAGE. The effect of Nb density on the binding and uptake of PEGylated liposomes was tested in MET-positive and MET-negative cancer cell lines by flow cytometry and confocal microscopy. The targeting functionality was investigated in a biologically relevant environment by pre-incubating NTLs and MET-TLs with high concentrations of fetal bovine serum (FBS) and human plasma before the uptake analysis. Finally, the interaction with circulating immune cells was investigated using an ex vivo human blood assay.

## 2. Results and Discussion

### 2.1. Characterization of Non-Targeted and MET-Targeted Liposomes

To investigate the impact of nanobody density on targeting efficiency, MET-TLs were prepared by the functionalization of the liposome surface with Nbs (15.6 kDa) targeting the MET receptor (MET-Nbs). These Nbs contain an engineered C-terminal cysteine that reacts with MAL-PEG-DSPE via maleimide-thiol click chemistry (Figure 1A). The conjugation of Nbs to NPs via a maleimide-thiol reaction has been previously employed by our group using an N-succinimidyl-S-acetyl-thioacetate (SATA) modification [17,42]. In this case, thiol groups are randomly introduced in the lysine residues, by virtue of which there is limited control over the stoichiometry. Moreover, Nb binding affinity might be affected when the lysines are present in the complementary determining regions (CDRs). On the contrary, site-directed conjugation guarantees that the Nbs retain their conformation and are tethered in the correct orientation for maximum exposure of the CDRs. Intein-mediated protein ligation was the first site-specific conjugation method used by our group to obtain Nb-targeted liposomes. However, this strategy requires several steps that must be tightly controlled for maximum conjugation efficiency [16]. To enable scaling up of the conjugation reaction for future clinical translation, we replaced intein with a C-terminal cysteine that only requires a reduction step prior to conjugation with maleimide-functionalized liposomes.

Different amounts of MET-Nbs were conjugated to the surface of liposomes by varying the amount of MET-Nbs added to the reaction mix. The degree of conjugation was determined by SDS-PAGE, where the MET-Nb-MAL-PEG-DSPE complex should yield a band at 18.5 kDa. Liposomes were first prepared with 8% of MAL-PEG-DSPE, which yielded two bands of a higher molecular weight than 15.6 kDa (Figure 1B, lanes 2 and 3). While the band at around 30 kDa corresponds to the MET-Nb dimer, the band at approximately 25 kDa suggests that the MET-Nbs reacted with more than one MAL-PEG-DSPE via the two cysteines from the internal disulfide bond of the MET-Nbs that is cleaved by TCEP. This phenomenon was less pronounced when the TCEP reduction conditions were optimized to decrease the amount of reduced internal disulfide bonds whilst keeping the conjugation efficiency close to 100% (Figure 1B, lanes 4 to 7). Decreasing the amount of MAL-PEG-DSPE but keeping the total amount of PEG-DSPE at a molar ratio of 8% also improved the conjugation of MET-Nbs to a maleimide group at more stoichiometric equivalence (Figure 1C). However, when 1% and 2% MAL-PEG-DSPE were used, a reduction in the overall conjugation efficiency was observed as evidenced by the remaining free Nb band, which entailed >50% in the case of MET-TLs prepared at the highest concentration of MET-Nbs (64 µM) (Figure 1C,D). These results demonstrate that a large number of conjugation sites are necessary to improve the conjugation efficiency. The combination of 4% MAL-PEG-DSPE with 4% PEG-DSPE resulted in the highest conjugation via the C-terminal cysteine (Figure 1C,D), and therefore this ratio was used in all subsequent experiments.

NTLs and six MET-TL formulations with a different MET-Nb density were successfully prepared by varying the molar ratio between MET-Nbs and MAL-PEG-DSPE from 1:500 to 1:12. As shown in Figure 2A, a band of higher molecular weight than the free MET-Nb was detected, as well as a proportional increase in the band intensity with the amount of MET-Nbs added to the reaction mix. Conjugation efficiency was lower for the highest Nb densities (Figure 2A), whilst unconjugated MET-Nbs were efficiently removed by dialysis (Figure 2B). For the NTLs (Figure 2A, lane 2), no MET-Nbs were detected as expected.

The average hydrodynamic diameter of the MET-TLs varied between 160 and 200 nm, with the 1:12 and 1:25 MET-TLs showing the greatest size and slightly greater size than the NTLs (Figure 2C). For all formulations, the PDI was below 0.2 and the ζ-potential was around −20 mV, which is in line with what has been reported for PEGylated liposomes [43,44]. The particle size determined by NTA revealed smaller liposomes when compared to the DLS measurements, as is expected given that the intensity of the scattered light is proportional to the sixth power of the particle diameter. The bigger nanoparticles, even in dispersion with a low size distribution, skew the average NP size towards higher values in DLS [45,46]. For this reason, NTA was used to determine the nanoparticle number per mL for the liposome stock before conjugation with MET-Nbs. This value, in combination with the conjugation efficiency, was subsequently used to estimate the number of MET-Nbs per liposome. Approximately 800 MET-Nbs per liposome were calculated for 1:12 MET-TLs and 40 for the 1:500 MET-TLs. These values can be converted to 1.9 and 0.1 MET-Nbs per 100 nm^2^, respectively (Figure 2C). Considering that Nbs usually have a diameter of 2.5 nm [47], a maximum of 20 MET-Nbs would fit in 100-nm^2^ plane (i.e., 9000 MET-Nbs per liposome). However, the number of MET-Nbs that can be conjugated to the liposomes is limited by the amount of MAL-PEG-DSPE and the steric hindrance imparted by the first MET-Nbs that react with maleimide. Transmission electron microscopy images (Figure 2D) revealed spherical liposomes for both NTLs and 1:50 MET-TLs, with dimensions in the same range as determined by DLS and NTA.

### 2.2. MET-Targeted Liposome Binding to and Association with Cells

The effect of MET-Nb density on the binding and association of MET-TLs was assessed using MET-positive (TOV+MET and TFK1) and MET-negative (TOV112D) tumor cell lines. The MET protein expression levels were confirmed by flow cytometry (Figure 3A).

MET-TLs bound more efficiently to TOV+MET cells than to TOV112D cells and also bound more efficiently to TFK1 cells than TOV+MET cells in a lipid concentration- and conjugation density-dependent fashion after 1 h of incubation at 4 °C. These data indicate a correlation between MET-TL binding and MET expression, as TFK1 cells showed the highest expression of MET receptors. In contrast, NTLs interacted very weakly with all cell lines. Altogether, these results confirm that the interaction of MET-TLs was mediated by the MET receptor (Figure 3B). Furthermore, a MET-Nb density-dependent increase in cell binding was observed in TOV+MET and TFK1 cells. However, 1:12 and 1:25 MET-TLs (the most densely decorated liposomes) also exhibited binding to TOV112D cells to a higher degree than that found for NTLs. This indicates that nonspecific binding occurs when the liposomes are overdecorated with Nbs, which should be prevented.

When net binding to TOV+MET cells was plotted by subtracting the TOV112D binding traces (Figure 3B), the 1:12, 1:25, 1:50, and 1:100 MET-TLs exhibited similar binding curves while the 1:250 and 1:500 MET-TLs were associated with lower binding affinity. These results demonstrate that the binding of MET-TLs to MET-expressing cells increases with increasing MET-Nb density, although there is an Nb density ceiling in terms of target selectivity.

Association studies were performed by incubating the cells with NTLs and MET-TLs for 5, 15, and 30 min at 37 °C. The association with the MET-negative (TOV112D) cells was very limited for both NTLs and the different MET-TLs (Figure 4A). For the cells that expressed MET, MET-TLs associated much more efficiently than NTLs (Figure 4A), which supports the previous findings that the association was mediated by MET (Figure 3B). The extent of association was dependent on the incubation time and Nb density. The association of 1:500 and 1:250 MET-TLs with TOV+MET and TFK1 cells was 2-to-3-fold lower than for the liposomes with higher Nb densities. The probability of the liposomes interacting with the MET-expressing cells is proportional to the Nb density on the liposome surface; thus, an increased MET-Nb density increases MET-TL avidity. However, a density above approximately 150 MET-Nbs per liposome only resulted in a slightly higher association. The association of the 1:25 MET-TLs with TOV+MET and TFK1 cells increased by 3% and 10%, respectively, compared to 1:50 MET-TLs, while for the 1:12 MET-TLs a slight reduction was observed. These results show that there is no advantage to having more Nbs per particle above a certain Nb density threshold (Figure 4A). Interestingly, TFK1 cells did not show a stronger association with MET-TLs than the TOV+MET cells, even though MET expression was higher for the TFK1 cells, and more profound binding was observed at 4 °C. These observations were made at roughly equal cell counts.

The specificity of the association of MET-TLs with MET-expressing cells was validated with a competition assay, where cells were pre-incubated with a saturating concentration of free MET-Nbs prior to their exposure to liposomes [35]. Under these conditions, the association of MET-TLs was greatly reduced to levels similar to those of NTLs (Figure 4B). The competition between the free MET-Nbs and MET-TLs was less evident for the 1:12 and 1:25 MET-TLs because these formulations had more non-specific interactions with the cells (Figure 4A), as also reflected by the saturation binding assay (Figure 3B).

The uptake of fluorescently labeled liposomes on MET-negative and MET-positive cells was assessed with confocal microscopy (Figure 4C,D). As shown in Figure 4C, no fluorescence was detected for the NTLs and the 1:25 and 1:250 MET-TLs following 1-h incubation at 37 °C, probably due to both the insufficient binding and low fluorescence quantum yield of the Cy5.5 fluorophore when excited with a 633 nm laser. At the 24 h timepoint, the high fluorescence signal was detected intracellularly in TOV+MET and TFK1 cells incubated with MET-TLs, whereas the signal remained low for NTLs, demonstrating that the MET-Nbs on the liposomal surface led to receptor-mediated endocytosis (Figure 4C,D). MET-positive cells incubated with 1:25 MET-TLs showed a higher intensity signal than that observed for 1:250 MET-TLs, which correlates well with the flow cytometry data (Figure 4A). However, 1:25 MET-TLs were also internalized by TOV112D (-MET) cells at a higher extent than 1:250 MET-TLs and NTLs (Figure 4D). As previously shown with flow cytometry, liposomes with the highest Nb densities induced non-specific binding and internalization of MET-TLs and therefore should not be considered for future studies.

Overall, from the data presented in Figure 3 and Figure 4, it was concluded that the most suitable range of MET-Nbs per liposome for optimal binding and specificity was from 150 to 300, which is equivalent to 0.4–0.8 MET-Nbs per 100 nm^2^ (Figure 2C). In line with these data, Yong et al. reported an optimal density of 4–12 anti-EGFR Nbs per Qdot [48]. This can be converted to approximately 0.3–1 Nb per 100 nm^2^, which is in accordance with the values reported here. On the other hand, in a previous study it was shown that liposomes conjugated with around 60 anti-EGFR Nbs per liposome induced an almost 2-fold decrease in EGFR expression compared to the ones with 30 anti-EGFR Nbs per liposome [42]. The anti-EGFR Nb density reported by Oliveira et al. corresponds to the 1:250 and 1:500 MET-TLs in this study, for which no major difference was observed with respect to cellular association (Figure 4A). Because the uptake of EGFR-targeted liposomes was not directly assessed in the study of Oliveira et al., it is difficult to compare the two formulations. However, the effect of the Nb density might additionally depend on factors such as the target type and cellular expression levels. For example, Woythe et al. showed that the uptake of silica NPs conjugated with cetuximab (anti-EGFR mAbs) depends on both the number of antibodies per NP and the EGFR expression on cells [49]. The ligand type can also influence the required ligand density for optimal binding efficiency. The ligand densities that were reported for NPs conjugated with mAbs, such as cetuximab and transtuzumab, vary between 100 and 1000 mAbs per NP, similar to the values tested for MET-Nbs. However, a high antibody density (more than 400 mAbs per NP) was necessary for efficient targeting and treatment efficacy [49,50]. These high densities might be required because of the random conjugation methods used, which decrease the number of functional antibodies available [49]. On the other hand, in another study it was reported that the cellular association of liposomes conjugated with anti-HER2 Fab fragments positively correlated with the ligand density and reached a plateau at approximately 40 Fab’ fragments per liposome [51], which is 3-fold lower than the values reported in the present study (Figure 2C). These findings indicate that if the targeting ligand has a high affinity towards the target, is stable, and is conjugated to NPs through site-specific conjugation methods, low-to-intermediate surface coverage suffices for efficient targeting.

### 2.3. Targeting Efficiency of MET-Targeting Liposomes in the Presence of Excess Proteins

To investigate whether MET-TLs can retain their targeting capacity after contact with plasma proteins, the MET-TLs responsible for the most pronounced association with MET-positive cells were pre-incubated with 50% *v*/*v* FBS or human plasma, as illustrated in Figure 5A. These protein concentrations have been previously tested in the context of the protein corona [33,52]. Usually, NP—protein corona complexes are separated from unbound proteins with size exclusion chromatography or ultracentrifugation. However, this purification step might perturb the protein corona [53]. It was therefore decided to directly add the formulations to the cells after the pre-incubation step. Moreover, given the high protein supplementation in the medium when compared to the standard culture conditions, the formulations were diluted to a final protein concentration of 10% or 50% *v*/*v*. This way, a possible effect provoked by blocking the receptors on the cells could be excluded. As shown in Figure 5B, the pre-incubation step with FBS affected the association of 1:25, 1:50, and 1:100 MET-TLs with both TOV+MET and TFK1 cells. However, the reduction in liposome association was always below 30% for all MET-TLs dispersed in the media with either 10% or 50% *v*/*v* FBS, suggesting that high protein concentrations do not impart substantial debilitating effects on target recognition by MET-TLs. Furthermore, increasing the MET-Nb density was not beneficial for overcoming the corona.

In a previous study, it was reported that the in vitro uptake of NPs was dependent on the source of medium supplementation [54]. Therefore, the association of MET-TLs was also studied in media supplemented with human plasma (either commercial or freshly isolated from the blood). As depicted in Figure 5C, the mean fluorescence intensity was not affected compared to the observed signal when TFK1 cells were incubated with MET-TLs and dispersed in FBS at the same protein concentration (Figure 5B), which indicates that human plasma does not further reduce receptor binding. As observed for the FBS-exposed samples, the association of MET-TLs was also reduced when the formulations were pre-incubated with 50% human plasma. Importantly, targeted liposomes still exhibited approximately a 20-fold higher association with TFK1 cells than NTLs. Interestingly, there was an increase from approximately 12% to 30% in the uptake when the cells were kept in 50% commercial human plasma during incubation with MET-TLs, while for the freshly isolated human plasma this was not verified.

The impact of the protein corona on targeting efficiency has been previously investigated for different targeting moieties. Similarly to our data, mAb-targeted liposomes and micelles retained their advantage over their non-targeted counterpart after exposure to plasma proteins. For anti-MUC-1 mAb-conjugated liposomes, there was a 2-fold decrease in the uptake, while for huA33-targeted micelles no differences in binding affinity were observed between the NPs incubated with protein free-medium and 100% human serum [55,56]. In contrast, transferrin- and Nb-functionalized silica NPs were found to lose their targeting capacity in culture media when they were supplemented with human serum [33,57]. The size of the targeting ligand was also reported to play a critical role in overcoming the protein corona. Polystyrene NPs coated with a peptide (DT7, <1 kDa) with an affinity towards the transferrin receptor showed an approximately 20% higher cellular uptake than those coated with transferrin (77 kDa) in protein-free medium, while after the in vivo protein corona formation no differences between the two targeting ligands were observed in vitro [58]. This might indicate that the protein corona has a higher impact on the targetability of smaller ligands. As the adsorption of proteins onto the NP surface depends on particle size, shape, and surface chemistry [59,60], it is important to systematically evaluate the targeting capacity of NPs in complex environments as it might give insights into their targeting efficiency in vivo.

### 2.4. Interaction of Non-Targeted and MET-Targeted Liposomes with Human Blood Cells

To assess the interaction of NTLs and MET-TLs with fresh human blood, an ex vivo assay using blood was employed as illustrated in Figure 6A. These experiments were performed independently with the blood from six healthy donors, whose white blood cell count is presented in Figure 6B. Human whole blood was incubated with 3 × 10^11^ particles/mL, as a meta-analysis performed by Ouyang et al. demonstrated that improved tumor accumulation was achieved when NPs were administered above a dose threshold of 10^12^ NPs per mouse [61]. This dose extrapolated to humans is 1.5 × 10^15^ NPs [62], which for an average adult with a blood volume of 5 L can be converted to 3 × 10^11^ particles per mL of blood. As observed in Figure 6C, NTLs and MET-TLs associated much more with granulocytes and monocytes than with lymphocytes, even though the latter represent 25–50% of the total white blood cell count. Less than 20% of lymphocytes stained positive for Cy5.5-labeled liposomes, while more than 60% of the granulocytes and monocytes were positive. Moreover, the MFI (i.e., the number of liposomes per cell) was highest for monocytes, followed by granulocytes and lymphocytes. These results are in agreement with other studies, where monocytes were found to interact to the largest extent with liposomes and micelles in both human and mouse blood [25,63,64]. Monocytes and granulocytes have the greatest capacity to remove particles from circulation due to their phagocytic activity. The interaction between NPs and these cells must therefore be minimized to increase NP circulation time and tumor accumulation.

Next, we aimed to understand the effect of Nb density on the association of MET-TLs with blood cells. Even though 1:500 MET-TLs have the lowest association with MET-positive cells, this formulation was included in the assay to test the hypothesis that a lower MET-Nb density translates to a reduced association of MET-TLs with immune cells. In Figure 6D, the data obtained for each formulation are compared using the Z-score and plotted per individual donor. The Z-score is a statistical method that represents the number of standard deviations a value is away from the mean of a group of values. For this study, if the Z-score of a sample is higher or lower than 0, then the MFI of that sample is above or below the average for all formulations, respectively. MET-TLs at 1:25 ratio showed the highest Z-score for both granulocytes and monocytes in five of the six donors, while NTLs and the other MET-TLs had a Z-score equal to or below 0. NTLs and 1:500 MET-TLs consistently showed the lowest association, and particularly for granulocytes a steady increase in MFI with increasing MET-TL density was observed (Figure 6C,D). However, only when the density of MET-Nbs was above approximately 300 MET-Nbs per particle did the association of MET-TLs with granulocytes and monocytes increase by roughly 2-fold (Figure 6C). These data suggest that, above a certain threshold, the MET-Nb density has a profound effect on the association of liposomes with granulocytes and monocytes. For the lymphocyte population, the opposite trend was observed; however, there was more variation between donors.

This ex vivo blood assay has been previously used to predict the clearance and tumor accumulation of antibody-targeted micelles in a mouse model [25]. The authors found that the micelles with a 50% or more antibody surface coverage had higher interactions with human monocytes, granulocytes, and dendritic cells than micelles with a low antibody density. Moreover, they observed that very densely decorated micelles had the highest clearance by the liver and spleen, whereas higher tumor accumulation was achieved for an intermediate density of the targeting ligand for tumors expressing the cognate receptor. In addition, targeted liposomes have previously been shown to have significantly lower blood residence times compared to the non-targeted nanocarriers but exhibited a comparable tumor accumulation [27]. These findings indicate that targeting ligands decrease the ‘stealth’ effects of PEG and thus induce faster clearance of NPs compared to untargeted NPs. However, as reported by Sivaram et al., adjusting the ligand density can restore the advantage of the targeted NPs by balancing the clearance rates and targeting efficiency. Even though tumor accumulation is mainly driven by the EPR effect, NPs conjugated with targeting ligands might be more effectively sequestered than non-targeted liposomes due to their higher affinity towards tumor cells. Based on the interaction of MET-TLs with immune cells, we speculate that MET-TLs with 150 to 300 Nbs per liposome will perform best in terms of circulation kinetics and tumor accumulation following intravenous administration in animal models. Such in vivo experiments are the focus of a separate study.

## 3. Materials and Methods

### 3.1. Materials

Phosphate buffered saline 10× solution (11.9 mM phosphate, 137 mM sodium chloride, 2.7 mM potassium chloride) was purchased from Fisher Bioreagents (Pittsburgh, PA, USA). DNAse I was obtained from Roche Diagnostics (Mannheim, Germany). One-mL CaptureSelect C-tagXL pre-packed columns were purchased from Thermo Scientific (Waltham, MA, USA) and HiTrap desalting columns were obtained from GE Healthcare (Chicago, IL, USA). Dipalmitoyl-*sn*-glycero-3-phosphocholine (DPPC), 1,2-distearoyl-*sn*-glycero-3-phosphoethanolamine-N-[methoxy(polyethyleneglycol)2000 (PEG-DSPE), maleimide PEG-DSPE, and Cy5.5-PE were obtained from Avanti Polar Lipids (Alabaster, AL, USA). Cholesterol, L-cysteine hydrochloride, and magnesium chloride were acquired from Sigma-Aldrich (St. Louis, MO, USA). Bolt 4–12%, Bis-Tris, 1.0 mm, mini protein gels, and Bolt MES SDS running buffer (20×) were obtained from Thermo Scientific. 1,4-Dithiothreitol (DTT) was acquired from Sigma Aldrich. Coomassie Blue PageBlue protein staining solution and PageRuler pre-stained protein ladder were obtained from Thermo Scientific. Roswell Park Memorial Institute-1640 (RPMI-1640) medium, Dulbecco’s modified Eagle medium (DMEM) high glucose, fetal bovine serum (FBS) (sterile filtered), trypsin EDTA solution (1×), antibiotic/antimycotic solution (100×), Dulbecco’s phosphate-buffered saline (PBS), and Tris(2-carboxyethyl)phosphine hydrochloride solution (TCEP) were purchased from Thermo Scientific. Vivaspin 2 100,000 MWCO (PES membrane) was obtained from Sartorius Stedim Lab (Gloucestershire, UK). All organic solvents were obtained from Biosolve (Valkenswaard, The Netherlands). Human plasma was obtained from Sigma-Aldrich and from healthy volunteers through the mini donor service of the University Medical Center Utrecht, the Netherlands. Donors provided informed consent. Institutional review board approval was waived for this route of whole blood curation.

### 3.2. Production and Purification of Nanobodies

MET-Nbs were produced and purified as described for other Nbs [65] with several procedural modifications. MET-Nbs were produced in BL21 *E. coli* transformed with a pET28 vector containing a C-terminal cysteine and an EPEA affinity purification tag. Single colonies were grown overnight in LB medium supplemented with 2% glucose and 100 μg/mL kanamycin. Overnight pre-culture (1:10 dilution) was expanded in a 5-L BioFlo/CelliGen 115 benchtop fermenter (Eppendorf, Hamburg, Germany) in a Terrific Broth medium containing 0.1% glucose and 100 μg/mL kanamycin. Once the log growth phase was reached, isopropyl β-D-1-thiogalactopyranoside (IPTG) at a final concentration of 1 mM was added to induce Nb expression. Bacteria were kept overnight at 25 °C. The next day, bacteria were centrifuged at 4800× *g* for 20 min at 4 °C, and the resulting pellet was resuspended in PBS and stored at −20 °C.

The periplasm fraction of *E. coli* was isolated with two cycles of freeze–thawing and two rounds of centrifugation at 4 °C (4000× *g* and 10,000× *g*, 20 min each). Periplasm was treated with DNAse and MgCl_2_ at a final concentration of 5 µg/mL and 25 mM, respectively, to reduce nucleic acid contamination and sample viscosity, and filtered through a 0.2-µm aPES membrane.

Nbs were purified by affinity chromatography using an ÄKTA Pure system (Cytiva Life Sciences, Marlborough, MA, USA) with a 1-mL CaptureSelect C-tagXL pre-packed column (Thermo Scientific). The column was equilibrated with PBS (pH = 7.4) and the periplasm fraction was applied to the column at a flow rate of 0.8 mL/min. The column was washed with 10 mL PBS, and subsequently a linear gradient elution with 10 mL of 20 mM Tris (pH = 7.0) and 2 M MgCl_2_ was applied at 0.5 mL/min. The Nb fractions were collected and buffer-exchanged to PBS (pH = 7.4) using 5-mL HiTrap desalting columns (GE Healthcare). Nb concentration was determined using absorbance at 280 nm (Nanodrop One, Thermo Scientific) and a molar extinction coefficient of 37,025 M^−1^·cm^−1^ (calculated with the ProtParam tool from Expasy, the Swiss Bioinformatics Resource Portal).

### 3.3. Cell Culture

The human extrahepatic bile duct carcinoma TFK-1 cell line (ACC 344) was purchased from Leibniz Institute DSMZ-German Collection of Microorganisms and Cell Cultures. The human ovarian carcinoma cell line TOV112D (CRL-11731) was obtained from the American Tissue Culture Collection (ATCC, LGC Standards, Wesel, Germany). TOV112D cells that stably express MET (here described as TOV+MET) were previously described [34]. TFK-1 cells were cultured in RPMI 1640 (cat. #11875093, Thermo Scientific), and TOV112D and TOV+MET cells were cultured in DMEM high glucose (cat. #11965, Thermo Scientific), both supplemented with 10% fetal bovine serum (FBS). TOV+MET cells were incubated with 250 µg/mL of zeocin (InvivoGen, San Diego, CA, USA) two days prior to an experiment to guarantee the presence of only MET-positive cells. The cells were maintained at 37 °C in the dark in a humidified atmosphere composed of 95% air and 5% CO_2_ (standard culture conditions).

### 3.4. MET Expression Assessment by Flow Cytometry

TFK1, TOV-112D, and TOV+MET cells (100,000 cells per sample) were fixed for 15 min with 4% paraformaldehyde (PFA), blocked with PBS containing 1% bovine serum albumin, and incubated with 50 µL of the primary antibody diluted 1:30 in blocking buffer (goat anti-human HGFR/c-MET antibody, cat. # AF276, R&D Systems, Minneapolis, MN, USA) for 45 min at room temperature (RT). The cells were washed twice and incubated with 50 µL of the secondary antibody diluted 1:200 in the blocking buffer (donkey anti-goat Alexa 488-labeled IgG, cat. #A11055, Invitrogen, Carlsbad, CA, USA) for 30 min at RT. Next, the cells were washed, resuspended in 100 µL of blocking buffer, and transferred to a 96-well plate. Fluorescence intensity was measured with a flow cytometer (Canto II, BD Biosciences, Franklin Lakes, NJ, USA) using the 488 nm laser line and a 530/30 nm filter. At least 10,000 events were collected per sample. Mean fluorescence intensity was calculated using FlowLogic Software (Mentone, Australia).

### 3.5. Preparation of (Fluorescent) Liposomes and Conjugation of MET-Nanobodies

The MET-TLs and NTLs were composed of 1,2-dipalmitoyl-*sn*-glycero-3-phosphocholine (DPPC), cholesterol, 1,2-distearoyl-*sn*-glycero-3-phosphoethanolamine-N-[methoxy(polyethyleneglycol)2000 (PEG-DSPE), and maleimide PEG2000-DSPE (MAL-PEG-DSPE) at a molar ratio of 77:15:4:4. During the optimization process to find the most suitable amount of MAL-PEG-DSPE, the fraction of PEG-DSPE and MAL-PEG-DSPE was kept at 8 mol%. Fluorescently labeled liposomes were prepared by adding 0.2 mol% of 18:0 Cy5.5-PE at the expense of DPPC. The liposomes were prepared by the lipid film hydration method as previously described [66]. In brief, lipids were mixed at predefined ratios in chloroform, transferred to a 100 mL round-bottom flask, and the solvent was evaporated in a rotary evaporator at 60 °C and kept under a nitrogen atmosphere for 30 min. The lipid film was hydrated with a physiological buffer composed of 0.8% (*w*/*v*) NaCl, 10 mM HEPES (pH = 7.4) to a final lipid concentration of 10 mM. Liposomes were extruded 2× through 400-nm polycarbonate membranes and 4× through 200- and 100-nm polycarbonate membranes using a 10-mL thermobarrel Lipex extruder (Northern Lipids, Burnaby, BC, Canada).

For the conjugation of the MET-Nbs onto the liposome surface, MET-Nbs in PBS were reduced with TCEP at an Nb:TCEP molar ratio of 1:200 and incubated at RT for 5 min. Next, TCEP was removed by buffer exchange to 0.8% (*w*/*v*) NaCl, 10 mM HEPES (pH = 7.4) with Zeba spin desalting columns (7 kDa MWCO, Thermo Scientific). MET-Nbs were mixed with liposomes at different ratios of Nb:MAL-PEG-DSPE and incubated in an orbital shaker for 2 h at RT and overnight at 4 °C. To quench the unreacted maleimide groups, a cysteine solution at 100× the molar concentration of MAL-PEG-DSPE prepared in 0.8% (*w*/*v*) NaCl, 10 mM HEPES (pH = 7.4) was added to the liposome dispersion (1:10 dilution) and left for 2 h at RT while rotating. NTLs were prepared by blocking the maleimide groups with cysteine as described for the MET-TLs. To remove non-conjugated and reacted MET-Nbs, the liposomes were dialyzed (Spectrum Spectra/Por Float-A-Lyzer G2 100 kDa, Spectrum Laboratories, San Francisco, CA, USA) against 0.8% (*w*/*v*) NaCl, 10 mM HEPES (pH = 7.4) for 16 h with 3× buffer exchange, followed by centrifugation at 3000× *g* with Vivaspin 100 kDa (Sartorius) to concentrate the liposome dispersion to the initial 10-mM lipid concentration.

### 3.6. Characterization of Non-Targeted Liposomes and MET-Targeted Liposomes

The hydrodynamic diameter and polydispersity of the liposomes were measured at a fixed scattering angle of 173° using a ZetaSizer Nano S (Malvern Instruments, Malvern, UK). The zeta potential was measured with a ZetaSizer Nano Z (Malvern Instruments). For both measurements, the liposomes were diluted 100× in 10 mM HEPES buffer (pH = 7.4).

The number of liposomes was determined by NTA. NTA measurements were performed using a NanoSight LM-10SH device (Malvern Instruments) equipped with a 532-nm laser. The liposomes were diluted 10,000-fold so that less than 100 particles were identified per frame. The liposomes were video-tracked with an EMCCD camera at RT. The average size was calculated with NanoSight 3.4 software from five measurements, each based on a 60-s video segment.

The size and morphology of the NTLs and MET-TLs were assessed by transmission electron microscopy (TEM) using a TFS Tecnai 20 microscope (FEI, Hillsboro, OR, USA) operated at 200 kV. For this analysis, a drop of the suspensions under study was placed on a carbon-coated copper grid for 2 min and stained with uranyl for 1 min.

The total lipid concentration was determined by an inorganic phosphate assay modified from [67]. The recovery efficiency was assumed equal for the lipids and cholesterol. To confirm MET-Nb conjugation to the liposomal surface, sodium dodecyl sulfate-polyacrylamide gel electrophoresis (SDS-PAGE) was performed. In short, the samples were combined with 4× SDS sample buffer (1:4 *v*/*v*, 30 µL total sample volume) and PBS and incubated at 90 °C for 10 min. After cooling down, the samples were loaded onto an SDS-PAGE gel (Bolt, 4–12% Bis-Tris Plus 1.0 mm × 10 wells, Invitrogen, Thermo Scientific) and run at 100 V for 50 min, employing 1 × 2-(N-morpholino)-ethanesulfonic acid (MES) buffer as running solution. Afterwards, the gel was stained with Coomassie blue (PageBlue protein staining solution) and scanned with a ChemiDoc Imaging System (BioRad, Hercules, CA, USA). Conjugation efficiency was determined by comparing the band intensity of free and conjugated Nbs using the Plot Lanes tool in ImageJ software (National Institutes of Health, Bethesda, MD, USA).

### 3.7. Nanobody Surface Density

The average number of MET-Nbs per liposome was calculated by dividing the total number of MET-Nb molecules per mL by the total number of liposomes per mL (before conjugation). The number of MET-Nbs per mL was determined by multiplying the number of moles by Avogadro’s constant and correcting for the conjugation efficiency (Section 3.6). The number of MET-Nbs per 100 nm^2^ was estimated by dividing the average number of MET-Nbs per liposome by the surface area of a liposome with an average radius of 122.8 nm (as determined by NTA, Section 3.6).

### 3.8. Binding of MET-Targeted Liposomes to Cells at 4 °C

TFK1, TOV-112D, or TOV+MET cells (5 × 10^4^ cells per sample) were added to a 96 round-bottom well plate and kept on ice for 15 min. Next, the cells were exposed to fluorescently labeled NTLs and MET-TLs serially diluted (from 500 µM starting lipid concentration and a dilution factor of 3) in RPMI containing 1% (*w*/*v*) BSA and 25 mM HEPES (pH = 7.4), and incubated for 1 h at 4 °C. After 3× washing to remove unbound liposomes and fixation with 4% PFA for 15 min, liposome binding was quantified by flow cytometry (Canto II) using the 633 nm laser line and a 780/60 filter. At least 10,000 events were collected in the gated region per sample. The mean fluorescence intensity was determined for each sample and corrected for the background signal intensity of non-treated cells using FlowLogic 8.3 Software.

### 3.9. Cell Association with and Uptake of MET-Targeted Liposomes

#### 3.9.1. Flow Cytometry

TFK-1, TOV-112D, and TOV+MET cells were seeded at a density of 3.5 × 10^4^, 3.0 × 10^5^, and 3.0 × 10^5^ cells per well, respectively, in a 48-well plate. The next day, cells were exposed to 60 µM of NTLs and MET-TLs (200 µL per well) for 5, 15, and 30 min at 37 °C in a humidified atmosphere with 5% CO_2_. A competition group with free Nbs was included in the assay. In this case, the cells were first incubated for 15 min with 1 µM of MET-Nbs, followed by either NTL or MET-TL incubation for 30 min under standard culture conditions. After washing thrice to remove unbound liposomes, the cells were detached with 100 µL of Accutase (cat. #A6964, Sigma Aldrich), transferred into a round-bottom well plate, and fixed with 4% PFA for 15 min. The liposome association with cells was measured by flow cytometry as described in Section 3.8.

#### 3.9.2. Confocal Microscopy

TFK-1, TOV-112D, and TOV+MET cells were seeded at a density of 2.0 × 10^4^, 1.2 × 10^4^, and 1.2 × 10^4^ cells per well in a black 96-well plate with a glass transparent bottom. The next day, NTLs and MET-TLs (500 µM lipid concentration was used to facilitate detection) were diluted in RPMI containing 1% BSA and 25 mM HEPES (pH = 7.4), which were added to cells and incubated for 1 h or 24 h at 37 °C under standard culture conditions. After 3× washing to remove unbound liposomes and fixation with 4% PFA for 20 min, the cells were stained with Hoechst 33342 (Thermo Scientific) for 20 min at 37 °C (1:750 dilution from 1 mg/mL stock in RPMI medium supplemented with 10% FBS). The cells were imaged with a Yokogawa High Content Imaging Platform (Model CV7000S, Yokogawa, Tokyo, Japan) using a 20× objective. Quantification of the fluorescence signal was performed using ImageJ and corrected for the background signal.

### 3.10. Cell Association with and Uptake of MET-Targeted Liposomes in the Presence of Excess Proteins

NTLs and MET-TLs were diluted in RPMI with 50% FBS or 50% human plasma (either commercial (cat. #P9523, Sigma-Aldrich) or obtained from fresh whole blood (Section 3.11)) to dispersion with a lipid concentration of 300 µM and incubated for 1 h at 37 °C using a thermomixer at 400 rpm. Next, the samples were diluted 5× in an RPMI medium supplemented with 50% FBS or human plasma and in a non-supplemented RPMI medium. The pre-incubated liposomes were added to cells (TFK1 and TOV+MET) and incubated for 30 min under standard culture conditions. After washing thrice to remove unbound liposomes, the cells were detached with 100 µL of Accutase, transferred to a round-bottom well plate, and fixed with 4% PFA for 15 min. Liposome association with cells was measured by flow cytometry, as detailed in Section 3.8.

### 3.11. Interaction of Non-Targeted Liposomes and MET-Targeted Liposomes with Whole Blood

Fresh human blood was collected from healthy subjects (*n* = 6) into citrate-anticoagulated Vacutainer tubes (BD Biosciences, Heidelberg, Germany). Complete whole blood counts were obtained using a CELL-DYN Emerald hematology analyzer (Abbott, Chicago, IL, USA). To obtain the plasma only, whole blood was centrifuged at 900× *g* for 15 min at RT. The plasma collected from the first centrifugation was again centrifuged at 950× *g* for 10 min at RT to remove any residual cells and stored at −20 °C.

Whole blood (180 µL) was incubated with NTLs and MET-TLs (3 × 10^11^ particles/mL) in a 2-mL Eppendorf tube for 1 h at 37 °C using a thermoshaker (400 rpm). Afterwards, red blood cells lysis buffer was added and the blood samples were incubated for 30 min on ice. Samples were washed twice with PBS (500× *g*, 7 min) and fixed with 1% PFA. Liposome association with immune cells was measured by flow cytometry as described in Section 3.8. Granulocytes, monocytes, and lymphocytes were gated based on forward and side scatter profiles, and the mean fluorescence emission intensity and percentage of the positive cells were quantified in the respective gated region using FlowLogic 8.3 Software. Data were corrected for non-treated whole blood.

## 4. Conclusions

In the present study, it was shown that the targeting efficiency of MET-TLs can be modulated by adjusting the MET-Nb density. The association of MET-TLs with MET-expressing tumor cells is an Nb density-dependent process; however, highly conjugated surfaces lead to a decrease in cell specificity. We also demonstrated that the targeting advantage of MET-TLs over the non-targeted counterpart is retained in protein-rich environments, indicating that MET-TLs can (partially) overcome the protein corona. The studies with ex vivo human blood showed that the MET-Nb density on liposomes affects immune cell interactions. Moreover, we confirmed that a panel of in vitro assays can be used to optimize formulations and, in part, steer clinical translation of nanomedicines. Future in vivo experiments will be conducted to verify the trends observed here and assess the drug delivery capacity of MET-TLs.

## Figures and Tables

**Figure 1 ijms-23-14974-f001:**
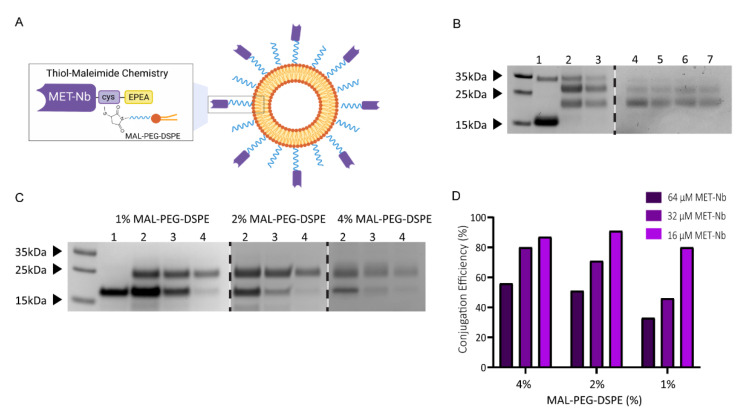
MET-Nb conjugation efficiency depends on TCEP reduction conditions and the percentage of MAL-PEG-DSPE and yields a different number of MET-Nbs per liposome. (**A**) Schematic of MET-targeted liposomes (MET-TLs) prepared via thiol-maleimide click chemistry. (**B**) Effect of the TCEP reduction conditions on MET-Nb conjugation efficiency to liposomes with 8% MAL-PEG-DSPE. SDS-PAGE with Coomassie blue staining: (1) free nanobody; (2) and (3) MET-TLs with 32 and 16 µM of MET-Nbs, respectively, and with MET-Nbs pre-treated with TCEP for 15 min at RT; (4) and (5) MET-TLs with 32 and 16 µM of MET-Nbs, respectively, and with MET-Nbs pre-treated with TCEP for 5 min at RT; (6) and (7) MET-TLs with 32 and 16 µM of MET-Nbs, respectively, and with MET-Nbs pre-treated with TCEP for 15 min on ice. (**C**) Effect of the percentage of MAL-PEG-DSPE on MET-Nb conjugation efficiency assessed by SDS-PAGE: (1) free nanobody; (2), (3), and (4) MET-TLs with 64, 32, and 16 µM of MET-Nbs, respectively, with 1, 2, and 4% of MAL-PEG-DSPE, respectively. (**D**) MET-Nb conjugation efficiency (%) at 1, 2, and 4% of MAL-PEG-DSPE estimated by comparing the band intensity of free and conjugated MET-Nbs using the Plot Lanes tool from ImageJ.

**Figure 2 ijms-23-14974-f002:**
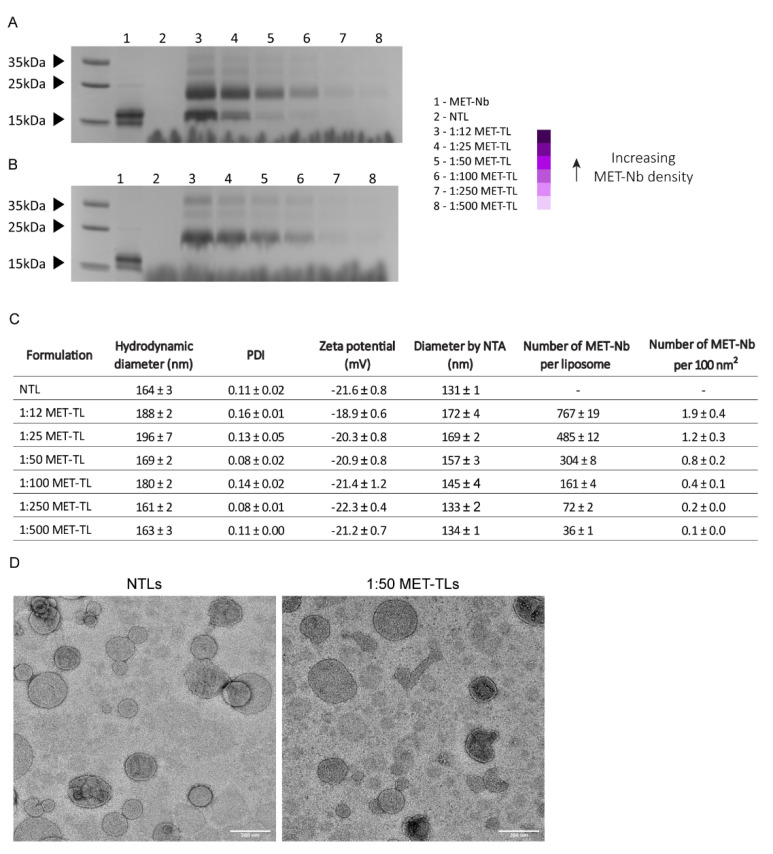
Liposomes with increasing MET-Nb density were prepared by varying the molar ratio between MET-Nbs and MAL-PEG-DSPE from 1:500 to 1:12. (**A**,**B**) SDS-PAGE with Coomassie blue staining: (1) free nanobody; (2), (3), (4), (5), (6), (7), and (8) non-targeted liposomes (NTLs), 1:12, 1:25, 1:50, 1:100, 1:250, and 1:500 MET-TLs, respectively, before (**A**) and after dialysis (**B**). (**C**) Physicochemical properties of NTLs and MET-TLs. Data represent mean ± SD (*n* = 3). (**D**) Transmission electron microscopy of NTLs and 1:50 MET-TLs. Scale bar represents 200 nm.

**Figure 3 ijms-23-14974-f003:**
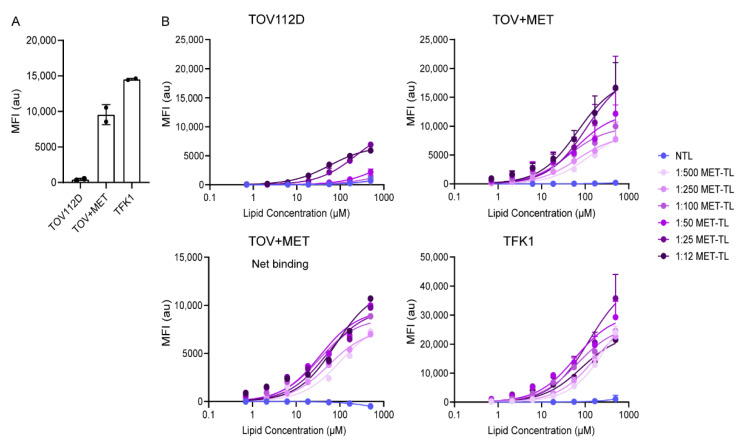
MET-targeted liposomes (MET-TLs) bind specifically to MET-expressing cells. (**A**) MET receptor expression on TFK1, TOV112D, and TOV+MET cells measured by flow cytometry. The mean fluorescence intensity (MFI) of the fluorescently labeled anti-MET antibodies is plotted as a function of cell line. (**B**) A binding experiment was performed by incubating TOV112D (-MET) and TOV+MET as well as TFK1 (+MET) cells with dispersions of non-targeted liposomes (NTLs) and MET-TLs with different MET-Nb densities for 1 h at 4 °C. The lipid concentration varied from 0.7 to 500 µM. Specific binding (net binding) to TOV+MET cells was inferred by subtracting the TOV112D binding traces, i.e., the negative control for MET, from the traces of MET-expressing cells. Data were normalized to the mean of the control group (non-treated cells) and expressed as mean ± SD of two independent experiments (performed in duplicate). The binding curves were fitted using GraphPad Prism.

**Figure 4 ijms-23-14974-f004:**
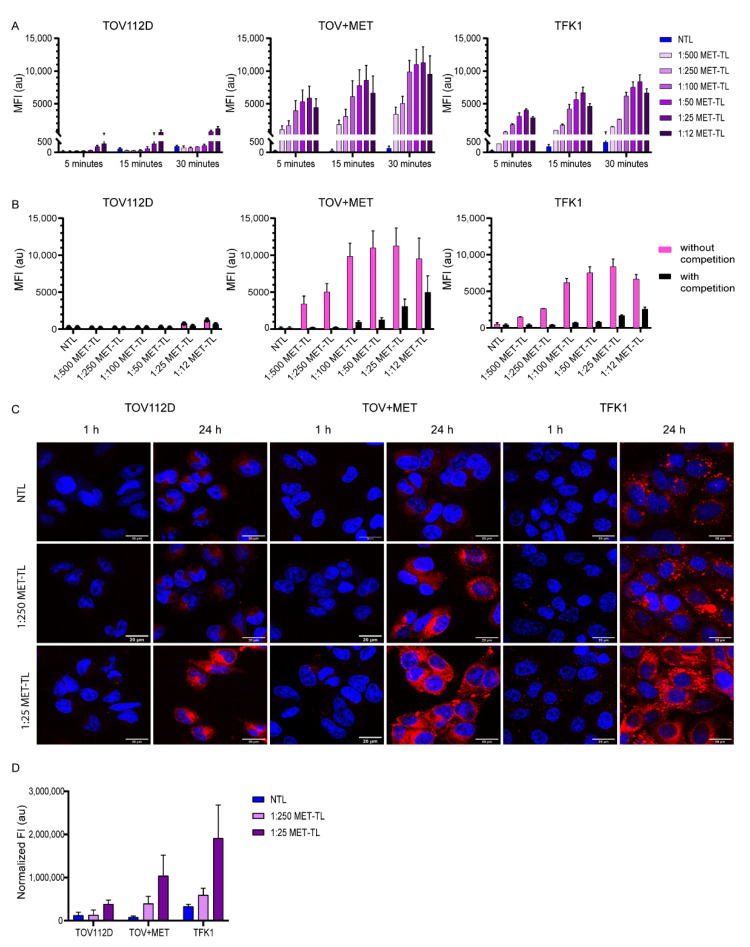
Association of MET-targeted liposomes (MET-TLs) with MET-expressing cells is mediated by the MET receptor and positively correlates with Nb density on the liposomal surface. (**A**) Cellular association with non-targeted liposomes (NTLs) and MET-TLs. TOV112D (-MET), TOV+MET, and TFK1 (+MET) cells were incubated with a dispersion of 60 µM of fluorescently labeled liposomes for 5, 15, and 30 min under standard culture conditions. (**B**) Specific association of MET-TLs with MET receptor. Cells were pre-incubated with a solution containing 1 µM of MET-Nbs for 15 min, followed by incubation with NTLs or MET-TLs. Liposome association in A and B was measured with flow cytometry and plotted as mean fluorescent intensity (MFI). Data were normalized to the mean of the control group (non-treated cells) and expressed as mean ± SD of two independent experiments (performed in duplicate). (**C**) Confocal microscopy of TOV112D, TOV+MET, and TFK1 cells incubated with a dispersion of 500 µM of fluorescently labeled liposomes for 1 and 24 h under standard culture conditions. Nuclei were stained with Hoechst 33342 (blue). Lipids labeled with Cy5.5 appear in red. Scale bar represents 20 µm. The confocal images clearly demonstrate uptake of the liposomes. (**D**) Quantification of the fluorescence intensity of NTLs and MET-TLs taken up by cells after 24 h incubation (based on confocal microscopy data).

**Figure 5 ijms-23-14974-f005:**
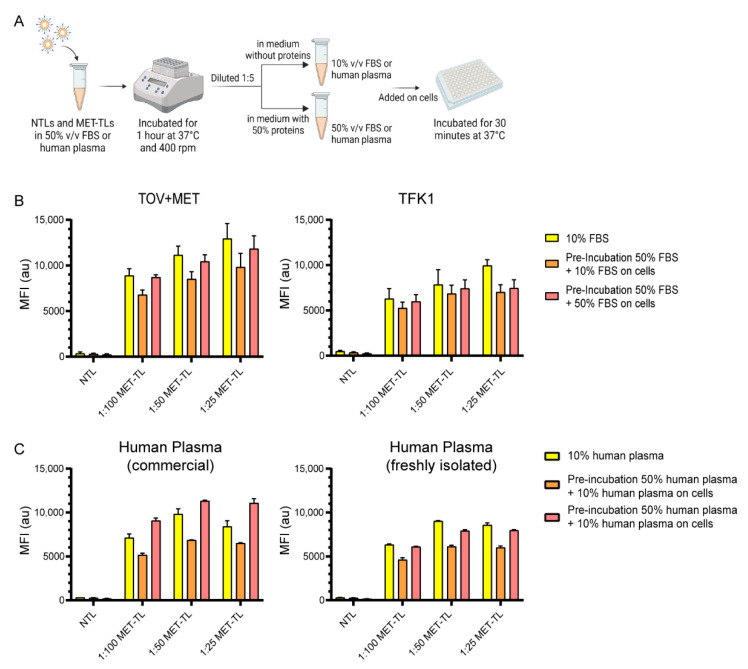
Effect of protein concentration on the targeting efficiency of MET-targeted liposomes (MET-TLs). (**A**) Schematic overview of the methodology employed to study the effect of pre-incubation on the MET-TLs with 50% FBS or 50% *v*/*v* human plasma on MET-TL-cell association. (**B**) Association of non-targeted liposomes (NTLs) and MET-TLs pre-incubated with 50% FBS with TOV+MET and TFK1 (+MET) cells. (**C**) Association of NTLs and MET-TLs pre-incubated with 50% human plasma with TFK1 cells. (**B**,**C**) Cells were incubated with dispersions of 60 µM of fluorescently labeled liposomes for 30 min under standard culture conditions. Liposome association was measured with flow cytometry and plotted as mean fluorescent intensity (MFI). Data were normalized to the control group (non-treated cells) mean and expressed as mean ± SD of two independent experiments (performed in duplicate).

**Figure 6 ijms-23-14974-f006:**
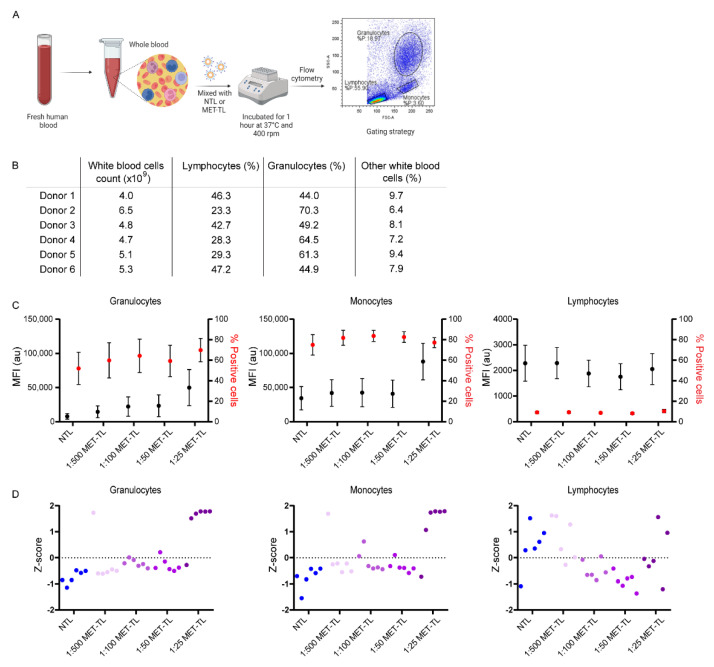
The effect of MET-Nb density on the interaction of MET-TLs with whole blood components. (**A**) Schematic of the method used to investigate the interaction of NTLs and MET-TLs with immune cells in whole blood. Fresh whole blood was exposed to a dispersion of 3 × 10^11^ NPs/mL and incubated for 1 h at 37 °C. Samples were analyzed with flow cytometry. Three different white blood cell populations (granulocytes, monocytes, and lymphocytes) were identified by their forward and side-scatter patterns. (**B**) Total count of white blood cells for each donor. (**C**) Mean fluorescence intensity (MFI) and percentage of Cy5.5-positive cells for granulocytes, monocytes, and lymphocytes. Data were normalized to the control group (non-treated whole blood) mean and expressed as mean ± SEM of six donors. (**D**) Z-score for each donor and formulation. Z-score was calculated based on the MFI of each NP and the standard deviation of the six donors.

## Data Availability

Not applicable.

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
