# Peer review of "The Impact of Nanobody Density on the Targeting Efficiency of PEGylated Liposomes"

_ijms, 2022, doi:10.3390/ijms232314974_

Round 1
Reviewer 1 Report
In this study, MET-Nb were conjugated to PEGylated liposomes at a density of 20 to 800 per liposome, and their targeting efficiency was evaluated in vitro. All of the results demonstrates that MET-TLs exhibited the highest association and specificity towards MET-expressing cells and retained their targeting capacity. Adjusting MET-Nb density can increase the specificity of NPs towards their intended cellular targets and reduce NP interaction with phagocytic cells. In addition, there are some problems in this manuscript that need to be explained, the specific opinions are as follows:
1. In page 12, line 433-435, the author presented “Based on the interaction of MET-TLs with immune cells, we expect MET-TLs with 150 to 300 Nbs per liposome likely perform best in terms of circulation kinetics and tumor accumulation following intravenous administration in animal models”. We think that the authors should supplement relevant animal experiments to prove this point.
2. In the section of “2.3. Targeting efficiency of MET-targeting liposomes in the presence of excess proteins (Page 8, line 313),Please the author supplement that the effect of protein concentration on the targeting efficiency of 1:500 MET-TLs, which consistent with section 2.4.
Author Response
In this study, MET-Nb were conjugated to PEGylated liposomes at a density of 20 to 800 per liposome, and their targeting efficiency was evaluated in vitro. All of the results demonstrates that MET-TLs exhibited the highest association and specificity towards MET-expressing cells and retained their targeting capacity. Adjusting MET-Nb density can increase the specificity of NPs towards their intended cellular targets and reduce NP interaction with phagocytic cells. In addition, there are some problems in this manuscript that need to be explained, the specific opinions are as follows:
Response: We thank the reviewer for the comments, which have helped us improve the manuscript. The comments are addressed below.
- In page 12, line 433-435, the author presented “Based on the interaction of MET-TLs with immune cells, we expect MET-TLs with 150 to 300 Nbs per liposome likely perform best in terms of circulation kinetics and tumor accumulation following intravenous administration in animal models”. We think that the authors should supplement relevant animal experiments to prove this point.
Response: In vivo experiments are underway and are conducted as part of a separate study. This is in line with our standard approach, where we establish broad in vitro proof-of-concept first (e.g., J Photochem Photobiol B. 2021 Mar;216:112146) and follow up with in vivo proof-of-concept (J Photochem Photobiol B. 2022 Sep;234:112500). To rightfully appease the reviewer, we have softened the language on page 12, line 442-445, as follows:
“Based on the interaction of MET-TLs with immune cells, we speculate that MET-TLs with 150 to 300 Nbs per liposome will perform best in terms of circulation kinetics and tumor accumulation following intravenous administration in animal models. Such in vivo experiments are the focus of a separate study.”
- In the section of “2.3. Targeting efficiency of MET-targeting liposomes in the presence of excess proteins (Page 8, line 313). Please the author supplement that the effect of protein concentration on the targeting efficiency of 1:500 MET-TLs, which consistent with section 2.4.
Response: In section 2.3, we investigated the 1:100, 1:50, and 1:25 MET-TLs because these were the formulations that showed the most profound association with MET-positive cells. In section 2.4, we were interested in determining whether the lower MET-Nb densities would perform better in case of interaction with immune cells. Because 1:500 MET-TLs did not outperform 1:100 MET-TLs, we concluded that it was not necessary to repeat the protein corona experiment described in section 2.3 with the 1:500 MET-TLs. To justify our decision, we added the following sentence in section 2.3 page 8 line 319-320:
“…the MET-TLs responsible for the most pronounced association with MET-positive cells…”
As for section 2.4 we have added the following text on page 10, line 398-400:
“Even though 1:500 MET-TLs have the lowest association with MET-positive cells, the formulation was included in this assay to test the hypothesis that a lower MET-Nb density translates to a reduced association of MET-TLs with immune cells.”
Reviewer 2 Report
The manuscript describes the liposomes' functionalization with nanobodies and the impact of these nanobodies density on the liposome surface. The in vitro studies showed that surface modification can improve selectively the cell uptake. Additionally, the amount of nanobodies in the liposomes impacts their biological behavior. The manuscript is very interesting and the results are promising, however, I would like to address some questions to the authors.
1- Introduction – What is the MET-Nb clinical implication? This point could be explored better in the introduction.
2- Material and Methods – Section 3.10: Please, if possible, add a reference or a rationale for the choice of concentration of FBS and human plasma used.
3- Results and Discussion – The statistical analysis of data shown in figures 2C, 3A, 4, 5B, 5C must be presented.
4- Conclusion – Will the behavior of MET-TLs in animal models be perform? Is it a future perspective?
Author Response
The manuscript describes the liposomes' functionalization with nanobodies and the impact of these nanobodies density on the liposome surface. The in vitro studies showed that surface modification can improve selectively the cell uptake. Additionally, the amount of nanobodies in the liposomes impacts their biological behavior. The manuscript is very interesting and the results are promising, however, I would like to address some questions to the authors.
Response: We thank the reviewer for the positive comments and the efforts to help us improve this manuscript. The questions are addressed below.
1- Introduction – What is the MET-Nb clinical implication? This point could be explored better in the introduction.
Response: We agree that more clarity must be given to the choice of MET receptor as target, therefore we added the following sentence on page 3 line 102-107:
“MET was chosen as target because it is a membrane receptor that is frequently overexpressed in numerous tumors, including breast cancer [36], cholangiocarcinoma [37], and glioblastoma [38]. Moreover, several anti-MET antibodies with inhibitory effect and anti-MET antibody-drug conjugates are undergoing clinical trials [39-41], which attests to the importance of exploring this protein as a target for drug delivery.”
2- Material and Methods – Section 3.10: Please, if possible, add a reference or a rationale for the choice of concentration of FBS and human plasma used.
Response: The concentration of FBS and human plasma used in this work have been previously tested in the context of protein corona, and more specifically its impact on ligand targeting capacity. Therefore, the following sentence was added with the respective references in section 2.3, page 8 line 321-322:
“These protein concentrations have been previously tested in the context of the protein corona [33, 52].’’
3- Results and Discussion – The statistical analysis of data shown in figures 2C, 3A, 4, 5B, 5C must be presented.
Response: Thank you for bringing this up; it is an important comment. We chose to not conduct any statistical analysis on our data sets for the following reasons. First, the samples sizes used (typically 2 samples in 2 replicate experiments; N = 4) renders the data sets unfit for statistical analysis. Even when one chooses to perform nonparametric analyses (because normality tests cannot be performed on small sample sizes, negating the possibility of using a parametric test), the outcomes will not be robust enough. This abrogates the utility of statistical testing in our case. Second, our R&D trajectory is predicated on establishing in vitro proof-of-concept (PoC) followed by validation studies using in vivo models. This approach was exemplified in the response to reviewer 1, see J Photochem Photobiol B. 2021 Mar;216:112146 (in vitro PoC) and the in vivo follow up in J Photochem Photobiol B. 2022 Sep;234:112500. For the in vitro development, optimization, and PoC studies we look at trends, whereas for certain key in vivo studies we power the groups such that outcomes are in fact statistically significant. Our aim is to develop therapeutics for human use, where test systems that are most remote from the human situation bear less weight than in vivo PoC studies that comprise part of the preclinical dossier (hence the lower sample size). Consequently, the non-statistical appraisal of trends is in our experience sufficient to establish a basis for more critical in vivo studies. Above all, the trends are rather unequivocal and thus the main conclusions can be drawn even without statistical analysis.
4- Conclusion – Will the behavior of MET-TLs in animal models be perform? Is it a future perspective?
Response: A follow-up in vivo study is planned and the results will be published as a separate study. For clarity, we included a sentence in the Conclusions section to clarify our intentions, page 16 line 658-659, which now reads:
“Future in vivo experiments will be conducted to verify the trends observed here and assess the drug-delivery capacity of such MET-TLs.”
Round 2
Reviewer 1 Report
This manuscript has introduced that the impact of MET-Nb density on the targeting efficiency of liposomes in vitro. Adjusting MET-Nb density can increase the specificity of NPs towards their intended cellular targets and reduce NP interaction with phagocytic cells. Compared with the previously reported MET targeting drug-delivery carrier, the MET-TLs developed by the author does not have much advantages in animal models, which is not conducive to the development of new drug-delivery carrier. After reading the entire article carefully, I regret to say that I think this paper is unsuitable for publication in International Journal of Molecular Science. In addition, I have some detailed comments on this manuscript.
1. We think about that the authors should supplement relevant animal experiments to prove this innovation and feasibility.
2. The physicochemical properties (such as size, shape, surface charge, surface polarization, hydrophilicity) of nanoparticles can affect the formation of protein crown, and the authors should supplement the physicochemical properties (shape, surface polarization, hydrophilicity, etc.) of liposomes with different MET-Nb densities.
Author Response
We thank this reviewer for the additional comments and as we introduced new data we submit the response as pdf.

Round 3
Reviewer 1 Report
After all consideration, we have decided to accept this article and publish it on IJMS.